# Improving the Mechanical Resistance of Hydroxyapatite/Chitosan Composite Materials Made of Nanofibers with Crystalline Preferential Orientation

**DOI:** 10.3390/ma15134718

**Published:** 2022-07-05

**Authors:** Ricardo Pascual Alanis-Gómez, Eric Mauricio Rivera-Muñoz, Gabriel Luna-Barcenas, José Rafael Alanis-Gómez, Rodrigo Velázquez-Castillo

**Affiliations:** 1División de Investigación y Posgrado, Facultad de Ingeniería, Universidad Autónoma de Querétaro, Querétaro 76010, Mexico; ricardo_pascualanis@hotmail.com (R.P.A.-G.); j.rafael.alanis@gmail.com (J.R.A.-G.); 2Centro de Física Aplicada y Tecnología Avanzada, Universidad Nacional Autónoma de México, A.P. 1-1010, Querétaro 76010, Mexico; emrivera@fata.unam.mx; 3Centro de Investigación y de Estudios Avanzados, Unidad Querétaro, Querétaro 76230, Mexico; gabriel.luna@cinvestav.mx; 4Escuela de Ingeniería, Universidad Anáhuac Querétaro, Querétaro 76246, Mexico

**Keywords:** nanotechnology, composite materials, crystal structure, bone-tissue implant

## Abstract

The stability and mechanical properties of hydroxyapatite (HAp)/Chitosan composite materials depend on the dispersion of HAp aggregates in the chitosan matrix and on the chemical interaction between them. Therefore, hexagonal cross-sectioned HAp nanofibers were produced using a microwave-assisted hydrothermal method. Glutamic acid was used to control the HAp crystal growth; thereby, nanofibers were obtained with a preferential crystalline orientation, and they were grown along the “c” axis of HAp crystal structures. This morphology exposed the (300) and (100) crystal planes on the surface, and several phosphate groups and calcium ions were also exposed; they were able to form numerous chemical interactions with the amine, hydroxyl, and carbonyl groups of chitosan. Consequently, the final mechanical resistance of the composite materials was synergistically increased. Nanofibers were mixed with commercial chitosan using a sonotrode to improve their dispersion within the biopolymer matrix and prevent migration. The HAp nanofiber/Chitosan composite materials showed higher mechanical resistance than that observed in similar materials with the same chemical composition that were made of commercial HAp powders, which were used as reference materials. The mechanical resistance under tension of the composite materials made of nanofibers was similar to that reported for cortical bone.

## 1. Introduction

Hydroxyapatite (HAp) is the main mineral component of human bone tissue. Synthetic HAp has a hexagonal crystal structure similar to that of natural HAp. The stoichiometry of synthetic HAp is (Ca_10_(PO_4_)_6_(OH)_2_), which differs from that of natural HAp because some calcium ions can be replaced by metal ions such as magnesium or sodium. These similarities in crystal structure and chemical composition make synthetic HAp an interesting material for use as a replacement in bone reconstruction or as an implant material for tooth restoration [1] due to its excellent biocompatibility, bioactivity, osteoinductivity, and osteoconductivity [2]; it also has a resistance to both ultraviolet and X-ray irradiations [3,4].

Various procedures have been used to synthesize HAp. In the sol–gel method, the precursors are blended in solutions, and the synthesis reaction occurs at room temperature and can last several days; the obtained product possesses high crystallinity [5,6,7]. Another study reported the use of solid-state reactions [8,9], but these reactions are difficult to control. None of these studies mentioned a control of the morphology, size, and structural parameters of HAp aggregates. Other authors have mentioned using different chemical substances (surfactant agents, amino acids, or halogen salts) in the synthesis reactions to control nucleation and crystal growth, which led to the obtained morphology of HAp assemblies [10,11,12,13].

The conventional hydrothermal method helps to obtain inorganic compounds with high crystallinity. A controlled heat supply produces steam from the aqueous solution, and the pressure increases within the reactor; subsequently, the synthesis reaction rate is also increased. This synthesis reaction takes a few hours [14,15,16,17]. The microwave-assisted hydrothermal method uses the energy provided by electromagnetic waves to start the chemical reaction, and the heat distribution is better in the whole reactor; consequently, the temperature is homogeneous in the chemical reactor. The latter produces a shorter reaction time, and the reaction ends in a few minutes. The tight control of crystal growth is also possible; the obtained product possesses high crystallinity and purity, and the morphology of the aggregates is easier to control [18,19,20,21,22,23].

Applications of HAp as a biomaterial have been possible owing to its above-mentioned properties. The chemical and thermodynamic stabilities of HAp and its response as an implant material for bone tissue are relevant characteristics to consider for good biomaterials. At the same time, chitosan is a promising biopolymer for biomedical applications attributable to its biocompatibility, biodegradability, antimicrobial properties, and minimal foreign-body reactions. Due to those characteristics, chitosan was selected for use in this research study.

The synthesis of HAp/chitosan composite materials has been studied by several authors, some of whom have used natural HAp for the elaboration of these materials [24,25,26], and others have made them with synthetic HAp [27,28,29]. Most of these authors have used these composites as biomaterials in the restoration of bone tissue, but other scientists have applied them in the removal of metal ions from water [30,31,32], as catalysts [33,34], and as drug releasers [35,36].

The homogeneous dispersion of HAp aggregates within the chitosan matrix and good blend stability are the main problems to be resolved. To achieve this, various methodologies have been proven by many authors. Some authors synthesized HAp nanocrystals in situ on a chitosan matrix [37,38], but HAp crystallinity was poor, and its morphology was not well-defined.

Other researchers have tried to obtain hydroxyapatite/chitosan composites by adding previously synthesized HAp nanostructures (mainly particles) to biopolymers with the additional purpose of improving biocompatibility, bioactivity, and bonding between constituents in the blend [39,40]. Most of the HAp crystals used in these materials were randomly grown, and the authors did not mention how well dispersed HAp aggregates were in chitosan. Moreover, several authors have focused their efforts on obtaining scaffolds made of HAp/chitosan composite materials [41,42,43,44]. In the elaboration of these supports, scientists were concerned with achieving an excellent chemical interaction between HAp and chitosan in the homogeneous dispersion of HAp aggregates and achieving adequate porosity that allows the diffusion of nutrients, ions, and cells required to carry out the bone tissue regeneration process to be obtained.

Considering the aforementioned, this work aimed to improve the tensile strength and blend stability of HAp/chitosan composite materials through a homogenous dispersion of HAp nanofibers inside the chitosan matrix and a strong chemical interaction between the inorganic and organic phases. To achieve these objectives, in the elaboration of HAp/chitosan composite materials, hexagonal cross-sectioned HAp nanofiber were specially synthesized by means of the microwave-assisted hydrothermal method (MAHM) using glutamic acid to guide crystal growth. MAHM helps to obtain HAp with high crystallinity and high purity. Glutamic acid strongly controlled the HAp crystal growth; therefore, a nanofiber morphology with a hexagonal cross-section was obtained. Nanofibers also had a remarkable preferential crystalline orientation in the [300], and they were grown along the “c” axis of HAp crystal structures. A sophisticated structural arrangement of nanofibers in the interior of the microfibers was obtained using a combination of MAHM and glutamic acid in the Hap synthesis process. These crystalline and structural characteristics of HAp contributed to the increase in the mechanical resistance under tension of the nanofibers, which influenced the final mechanical properties of the composite materials.

This specially designed morphology of nanofibers with a hexagonal cross-section left the (100) and (300) planes of the HAp crystal structure exposed on the lateral fiber surfaces [2]. Several phosphate groups and calcium ions were present in these crystalline planes; as a result, they were also exposed on the lateral surfaces of the fiber. The availability of phosphate and calcium ions enabled the formation of numerous chemical interactions with the amine, hydroxyl, ether, and carbonyl groups of chitosan, which generated a synergistic increase in the final mechanical resistance and higher structural stability of the composite materials. Another novel contribution of the present study is the use of a sonotrode to improve the dispersion of HAp nanofibers in the biopolymer matrix. Better dispersion prevented the migration of nanofibers and with the possibility of forming a higher number of chemical interactions resulted in the greater mechanical resistance and structural stability of the composite material.

The concentration of nanofibers in the composite materials was varied to analyze their influence on the final mechanical resistance of the composite material under tension. Concentrations of 5%, 10%, and 20% by weight (based on the chitosan content) were used separately. Similar materials were also obtained using commercial HAp powders rather than nanofibers, and these materials were used as references in characterization studies.

## 2. Materials and Methods

### 2.1. Synthesis of Materials

HAp nanofibers were synthesized via a microwave-assisted hydrothermal method using calcium nitrate (CaNO_3_), potassium phosphate dibasic (K_2_HPO_4_), and potassium hydroxide (KOH) as precursors and glutamic acid to guide crystal growth. Fine points of the synthesis process and the growth of nanofibers have been described in previous works [45,46,47].

The synthesis procedure was developed as follows: First, a glutamic acid [C_5_H_9_NO_4_-H_2_O] (J.T. Baker FW 147.13; Phillipsburg, NJ, USA) and Calcium Nitrate Tetra hydrate [Ca(NO_3_)_2_-4H_2_O] (Golden Bell FW 236.16; Bell, CA, USA) solution was prepared in 200 mL of deionized water. This solution was heated and magnetically stirred for approximately two hours until the complete dissolution of the solutes was achieved. Alternatively, another dissolution of monobasic potassium phosphate [KH_2_PO_4_] (Mallinckrodt Chemicals FW 136.09; Chesterfield, UK) along with potassium hydroxide [KOH] (Sigma-Aldrich 221473; St. Louis, MO, USA) was performed in 200 mL of deionized water. Vigorous agitation was used to dissolve the chemical substances. When both solutions were prepared, they were mixed to form the reaction mixture, and agitation was continued. Subsequently, this reaction mixture was transferred to eight quartz vials (each containing 50 mL), and these vials were placed inside a Shyntos 300 (Anton Paar, Ashland, VA, USA) microwave oven. The synthesis reaction was carried out at 170 °C, with a heating time of 10 min (Since 25 °C), a reaction time of 45 min, and a cooling time of 10 min.

When the synthesis reaction was complete, HAp was deposited at the bottom of the vials, and the precipitate was filtered and washed repeatedly with an isopropanol–water mixture. The final product was then collected and dried.

The synthesized HAp nanofibers were then used to obtain chitosan–HAp composites, which were prepared using commercial chitosan powders (Sigma Aldrich, St. Louis, MO, USA; molecular weight range of 190,000–310,000). A total of 1 g of chitosan was dissolved in 99 mL of an acetic acid solution (1% by weight) and 4 mL of a buffer solution (NaH_2_PO_4_/Na_2_HPO_4_) 0.1 M to keep pH = 7.2, and this dissolution was stirred for six hours. Subsequently, HAp nanofibers were added to the solution at separate concentrations of 5%, 10%, or 20% by weight (based on chitosan), and a Hielsher model UP200Ht sonotrode was used to stir the mix in order to enhance the dispersion of the HAp fibers inside the chitosan matrix. Agitation was performed by applying a series of one 30 s pulse of ultrasonic wave emission, followed by a one 10 s period of relaxation. All the stirring processes included 30 series. Agitation was applied in series to avoid structural damage to the HAp nanofibers and chitosan macromolecules. Finally, the composite mix was divided and placed in glass Petri dishes to evaporate the solvent and obtain the films. Similar materials were prepared using commercial HAp powder instead of nanofibers and were used as reference materials in the characterization procedure. The sample containing 5% wt. of commercial HAp was called “Reference 1”; the composite material with 10% wt. of commercial HAp powders was called “Reference 2”; finally, the composite material made with 20% wt. of commercial HAp was called “Reference 3”.

### 2.2. Characterization of HAp Nanofibers

#### 2.2.1. X-ray Diffraction

To identify the crystalline phases and determine the possible preferential crystalline orientation, powder X-ray diffraction was used to analyze the HAp nanofibers [48,49]. A D8 Advance diffractometer, built and designed by Bruker (Karlsruhe, Germany), was used to perform crystal structure analyses. An accelerating voltage of 30 kV and a current of 30 mA were used to produce Cu Kα radiation (0.15406 nm wavelength), and all diffraction experiments were carried out from 10° to 80° on a 2θ scale with an angular step of 0.05°. No milling process was performed on any sample.

#### 2.2.2. Scanning Electron Microscope

The morphology and topology of the HAp nanofibers were observed by means of two scanning electron microscopes, where one of them was JEOL JSM-6060 LV (Jeol, Akishima, Tokyo, Japan) and the other was Hitachi SU8200 (Hitachi High-Technologies, Tokyo, Japan). Accelerating voltages of 10 kV and 20 kV were applied to perform the observations, and secondary electrons were used to form all images. Dry samples were collected directly from the synthesis process, placed on a sample holder, and glued by carbon paint [50,51].

#### 2.2.3. High-Resolution Electron Microscopy

The nanofiber morphology and crystalline structure of the HAp were analyzed using a transmission electron microscope, JEOL JEM 2010 HT (Jeol, Akishima, Tokyo, Japan), with a resolution of 0.19 nm. The samples were placed on lacey carbon films on 200-mesh copper grids. An accelerating voltage of 200 kV was used during all observations, and high-resolution images of the samples were obtained. Selected-area electron diffraction (SAED) was also performed whenever possible [50,52]. High-resolution images were analyzed using Digital Micrograph software (Gatan, Ametek, CA, USA), and fast Fourier transforms (FFTs) were generated to analyze the crystal structure of HAp.

### 2.3. Characterization of Composite Materials

#### 2.3.1. X-ray Diffraction

X-ray diffraction analyses were performed on all types of composite materials, those made of commercial HAp (references) and the others prepared with HAp nanofibers. As mentioned, the HAp concentrations separately used in both cases were of 5%, 10%, and 20% by weight. These XRD analyses were performed using the same equipment used to characterize the HAp nanofibers under the aforementioned conditions [37,53].

#### 2.3.2. Scanning Electron Microscopy

Observation through (SEM) observations was carried out for all the composite materials using the same electron microscopes previously mentioned [25,37,54]. In this study, the topology of composite materials was analyzed, and the microstructure of commercial HAp particles or HAp nanofibers was examined. Moreover, the interfaces between HAp and chitosan were studied, emphasizing the observation of the adhesion between the blend components and evaluating the dispersion of HAp aggregates in the chitosan matrix.

#### 2.3.3. Fourier-Transform Infrared Spectroscopy (FTIR)

Pure HAp powders, pure HAp nanofibers, pure chitosan, and composite materials were analyzed using a Bruker Vectro 33 spectrometer (Bruker, Billerica, MA, USA) to analyze the chemical structure of all materials and determine the chemical interactions between the organic and inorganic phases in the composite materials [54,55,56]. The thin films were analyzed using the transmission technique (except for the two types of pure HAp, which were analyzed by diffuse transmittance). All the film samples possessed similar thicknesses and were weighed before the analysis. The analyzed samples (including HAp powders and HAp nanofibers) had the same mass quantity (20 mg), and the acquired spectra were the result of 50 scans performed on each sample and recorded with a wavenumber resolution of 4 cm^−1^.

#### 2.3.4. Raman Spectroscopy

Raman analyses were performed using a Bruker Senterra spectrometer (Bruker, Billerica, MA, USA) to complement the chemical structure analyses of all composite materials. The operating conditions during all analyses were a voltage of 100 mV, a wavenumber resolution of 3–5 cm^−1^, and a 20× objective. The samples were then milled and weighed for further analysis. Once again, 20 mg samples were collected. They were used for the analyses [27,57].

#### 2.3.5. Mechanical Properties

The mechanical resistance under tension was determined for all composite materials using a universal mechanical testing machine, Zwick/Roell Z005, (Zwick Roell, Ulm, Germany) with a load cell of 5 kN. All the tensile tests were performed according to the ASTM D638 standard. Specimen preparation was performed based on this document. All the specimens were obtained using a cutter with the shape and dimensions indicated in the standard. The cut was performed carefully so as to avoid generating stress within the specimen as much as possible. Tensile tests were carried out using a speed in the moving crosshead of 1 mm/min and a temperature of 25 °C [41,56]. Similar tests were also performed on the reference composite materials, and the results were compared.

After the tensile tests, the fracture surfaces of the nanocomposite specimens were observed using a Hitachi SU8200 scanning electron microscope.

## 3. Results

### 3.1. Characterization of HAp Nanofibers

#### 3.1.1. X-ray Diffraction (XRD)

Initially, commercial HAp powders were analyzed by XRD, and the characteristic diffractogram of this type of HAp is displayed in Figure 1a. This diffractogram shows a noisy baseline and wide Bragg reflections. The width of these Bragg reflections indicates a reduced crystallinity in the sample. Crystalline-phase identification was performed by comparison using the powder diffraction file (PDF) data bank provided by the International Center for Diffraction Data (ICDD), and the only crystalline phase found was hydroxyapatite according to PDF # 09-432. The positions and intensities of all Bragg reflections in the diffractogram shown in Figure 1a agree with the data contained in the PDF, and the most intense Bragg reflection was produced by (211). The average crystallite size was calculated using the Scherrer equation, and the determined value was 6.57 nm.

HAp nanofiber samples were also analyzed, and their respective diffractograms showed significant differences when compared with those obtained from commercial HAp samples. A typical diffractogram of HAp nanofibers is shown in Figure 1b. A single crystalline phase was identified using PDF # 09-432. The presence of only one crystalline phase is indicative of high purity. The positions of all Bragg reflections in the diffractogram were as expected, but the intensities of some of them differed from those listed in the PDF. If the intensity of the Bragg reflection produced by (211) is considered 100% as in the mentioned PDF, the most intense Bragg reflection in the diffractogram was produced by (300), and this reflection had an intensity three times larger than that registered by (211). The signal corresponding to the (100) plane also showed a significant increase in its intensity. According to # 09-432, this Bragg reflection should have an intensity of only 17.7%. However, the registered intensity of this particular reflection in the diffractograms was 75% of the intensity determined for (211). In contrast, the Bragg reflection produced by (002) underwent a considerable intensity reduction, and the PDF indicates an intensity of this reflection of 35.9%; however, the intensity of the Bragg reflection produced by (002) in the diffractogram of HAp nanofibers was almost null. The variations in the intensity of the aforementioned Bragg reflections are indicative of a remarkably preferential crystalline orientation in [300]. Similar results were found in previous works [45,46,47]. Moreover, the diffractogram displayed in Figure 1b shows thin and well-defined Bragg reflections with a smooth baseline. A narrow reflection width is associated with high crystallinity.

Another notable difference between the diffractograms in Figure 1a,b is the intensity of all the Bragg reflections. The diffractogram of the commercial HAp powders had a maximum intensity of 2460 (a.u.) of the reflection produced by (211). In contrast, the diffractogram of the HAp nanofibers reached a maximum intensity of 13,150 (a.u.) of the (300) reflection. This difference in intensity could have been related to the high crystallinity.

The difference in crystallinity between the commercial sample and the HAp nanofibers was quite noticeable. The crystallinity of the HAp nanofibers was the result of the use of the microwave-assisted hydrothermal method in the synthesis reactions, and the preferential crystalline orientation is a consequence of the use of glutamic acid to control crystal growth.

#### 3.1.2. Scanning Electron Microscopy (SEM)

Commercial HAp powders and HAp nanofibers were observed using SEM, and notable differences in the morphology and topology were observed when the two types of samples were compared. Figure 2a shows an SEM micrograph of a commercial HAp sample, and its morphology was analyzed. Bacillus-like irregular particles were observed in this sample. The particle surfaces appeared to be smooth, and the average particle length was 4 μm, while crystalline facets were not clearly observed; probably, different crystalline planes were exposed on the particle surfaces. The latter corroborates the low crystallinity found by XRD for this HAp sample.

On the other hand, the HAp synthesized by our research group showed a morphology of microfibers with a hexagonal cross-section and well-defined edges and lateral facets, as shown in Figure 2b. In the inset, it is possible to see how each microfiber was made by several nanofibers. The lateral facets in all the microfibers were quite smooth, and the (100) and (300) planes were exposed on their surfaces. Microfibers had an average maximal diameter of 4.52 μm and a length of 43 μm. Finally, the lateral facets of the microfibers had large surface areas.

The analysis of this particular arrangement of nanofibers inside microfibers is discussed in detail in previous works [43,44,45]. The morphology of the fibers is the result of the use of glutamic acid to guide HAp crystal growth.

#### 3.1.3. High-Resolution Transmission Electron Microscopy (HRTEM)

HRTEM observations of the two types of HAp samples also revealed significant differences. A typical TEM micrograph of a commercial HAp sample is shown in Figure 3a. Once again, it was possible to observe small irregular particles of different sizes, and no particular morphology was observed. In Figure 3b, a fast Fourier transform (FFT) was produced using the micrograph region inside the white rectangle. The FFT was generated using digital micrograph software designed by Gatan. As a result, a ring pattern was obtained, which was indicative of the contribution of several small crystals to pattern formation. FFT was indexed, and the contribution of the (211), (112), (300), and (202) planes to the generation of the ring pattern was identified, which corroborated the results found by XRD.

In the case of HAp nanofibers, HRTEM observations confirmed the arrangement of nanofibers inside the microfibers, as observed by SEM. In general, nanofibers also show hexagonal cross-sections and well-defined edges and facets [43,44,45], and they have an average maximal diameter of 76.4 nm. The high-resolution micrographs of the two nanofibers are displayed in Figure 4a,c, and an arrangement of lines and dots can be seen in the interior of the micrographs. Using the micrograph displayed in Figure 4a, an FFT was generated for the image region inside the white rectangle, and this FFT is shown in Figure 4b.

A dot pattern was obtained, which possessed two-fold rotational symmetry. This indicated that the lateral facets of the nanofibers received the electron beam in the orthogonal direction. The dot definitions and their arrangement within the pattern are indicative of high crystallinity. The FFT was indexed, and the contributions of the (110) and (002) planes were determined. The micrograph portrayed in Figure 4c shows a more evident arrangement of lines and dots within the nanofibers, which was also processed using digital micrograph software, and interplanar distances of 0.342 nm (associated with (002)) and 0.271 nm (related to (300)) were determined. The micrograph region within the white rectangle in this figure produced the FFT displayed in Figure 4d. Once again, the generated dot pattern had a two-fold symmetry, and the dot definition indicated high crystallinity. The contributions of the (002), (100), and (300) planes to form the dot pattern were identified.

The two-fold symmetry found in both FFTs was produced by the lateral facets of the nanofibers. As can be observed in the SEM micrographs, the hexagonal cross-section of the fibers is delimited by rectangular lateral facets. SEM and HRTEM results, along with those obtained from XRD, revealed that the (300) and (100) planes formed lateral facets, and consequently, these crystalline planes were completely exposed on the side surfaces of the nanofibers; due to their length, these lateral facets had a large area. Moreover, in both FFTs depicted in Figure 4b,d, it can be seen that the (002) planes were stacked along the fiber length, which is evidence that the nanofibers grew along the [001] direction.

### 3.2. Characterization of Composite Material

As previously mentioned, different chitosan–HAp composite materials were prepared. Concentrations of 5%, 10%, and 20% by weight of HAp nanofibers (related to chitosan) were used separately. Additionally, other composite materials with the same chemical composition were prepared using commercial HAp powders instead of nanofibers, and these were used as reference materials (References 1, 2, and 3, respectively) in the characterization procedure.

#### 3.2.1. X-ray Diffraction

The diffractograms in Figure 5a were obtained for a composite material made of commercial HAp powders. The three diffractograms in this figure were formed with the contribution of the chitosan matrix, which is an amorphous material that produced a wide signal around 21° in 2θ, and the HAp crystal structure, which generated Bragg reflections (PDF #09-432). The contribution of the amorphous phase in the diffractogram caused the hiding of the Bragg reflections produced by HAp, which were already poor in intensity owing to the low crystallinity of the powders. This overlap of reflections was more remarkable for the composite material containing 10% by weight of commercial HAp, and this material also showed the most intense signal generated by the amorphous matrix. The latter could result from a good mix of both components in the material.

In the case of the composite material made with HAp nanofibers, the diffractograms displayed in Figure 5b also show the contribution of both components in the mixture; however, the intensity of the Bragg reflections produced by the HAp crystal structure was notably more significant than that of the broad signal generated by the amorphous chitosan matrix, and the intensity of the Bragg reflection caused by the (300) plane was still notable. The crystallinity of HAp did not decrease, indicating that the process of elaborating the composite materials did not alter its crystal structure.

The latter is a consequence of the high crystallinity of the HAp nanofibers. Similar to the reference materials, once again, it was found that the contribution of the chitosan matrix was more significant for the composite material containing 10% by weight of nanofibers, which indicated that this composition allowed a good mix of both components to be obtained.

#### 3.2.2. Scanning Electron Microscopy

SEM observations were performed for the reference materials. In Figure 6a–c, it is possible to see the secondary-electron micrographs of the composite material surfaces, which have different HAp contents of 5%, 10%, and 20% by weight, respectively. In all these materials, an irregular topology was observed, and some small white dots appeared dispersed on the surface. These dots represented the commercial HAp particles. A good dispersion of HAp particles was achieved as a consequence of using a sonotrode to mix the components in the material. In all cases, most of the HAp particles were well coated with the chitosan film. Adhesion between the HAp particles and the chitosan matrix seemed good, which could be favorable for the mechanical properties of the composite materials.

The micrographs in Figure 6a–c showed a similar topology for all composite materials, and the number of particles near the sample surface seemed comparable in the materials with HAp concentrations of 10% and 20% by weight. The material containing 5% by weight of HAp commercial powder seemed to have a more significant number of particles on its surface, which could be a consequence of better particle dispersion.

In the case of composite materials made of HAp nanofibers, their topology appeared to be smoother (Figure 6d–f). Adhesion between the fibers and the chitosan matrix seemed quite good, and because of the use of a sonotrode, the dispersion of fibers within the matrix was quite good. Similar to the reference materials, most of the nanofibers were completely coated by the chitosan film, and most looked undamaged. They did not undergo fractures or breaks during the synthesis process of the composite materials. This result is relevant to the final mechanical properties of these composite materials and shows that the synthesis process did not significantly damage the structure of the nanofibers.

#### 3.2.3. Fourier-Transform Infrared Spectroscopy (FTIR)

The analyses of different composite materials made of commercial HAp revealed interesting results. Figure 7a,b show the FTIR spectra of pure commercial HAp, pure chitosan, and the different reference materials. The spectra showed several variations in their bands, but the most relevant changes were the following: the bands next to 1103 and 1028 cm^−1^ were produced by the phosphate group in the pure HAp, and the bands at 1063 and 1048 cm^−1^ were related to the vibration of the C-O bond in the glucose ring of pure chitosan [58,59] (Figure 7a). These bands likely experienced an overlap when the composite materials were formed, but something interesting in the spectra shown in Figure 7a is that the combined band not only decreased its transmittance, as expected, but its position was shifted from 1028 cm^−1^ in pure HAp to 1022 cm^−1^ in Reference 1 or up to 1024 cm^−1^ in Reference 2 and Reference 3. Finally, the phosphate groups in HAp had chemical interactions that widened the band and moved it to lower wavenumbers. The transmittance of this new band varied significantly, from the value registered for Reference 1 to that observed for Reference 2; however, this variation in the transmittance value was less evident from Reference 2 to Reference 3. This irregular variation in the transmittance of this band could be a consequence of the different numbers of chemical interactions generated by the phosphate groups. Other small bands at 960 cm^−1^ related to phosphate groups varied considerably in transmittance, and this modification was not consistent with the simple variation in the HAp concentration of the composite materials. The latter could also indicate the variable number of chemical interactions produced by the phosphate groups.

Moreover, Figure 7b displays the band registered at 1640 cm^−1^ related to the C=O bond of the amide groups, which indicated that the commercial chitosan used in this study was not fully deacetylated, and the last was corroborated by the peak at 1315 cm^−1^, which was also produced by the same carbonyl group [47]. In addition, the peak at 1545 cm^−1^ was attributed to the vibrations of the N-H bonds of the amine group [48]. The bands did not show any widening or shift, but the transmittance variation was not as expected when the concentration of HAp was increased in the composite materials. In the case of Reference 1, the mentioned bands showed a high transmittance. Subsequently, when the HAp concentration was increased up to 10% by weight in Reference 2, the transmittance of these bands significantly decreased, even though these bands had transmittance similar to those determined for the pure chitosan sample (dashed line). However, when the HAp content was increased again to 20% by weight in Reference 3, the transmittance of the bands at 1640, 1315, and 1545 cm^−1^ did not show a decrement; conversely, they showed an increment of approximately 6% compared with the transmittance observed for the sample with 10% by weight (Reference 2). This inconsistent variation in the transmittance of these signals could result from the chemical interactions between the amino and carbonyl groups. In addition, transmittance variations were observed in the peaks at 1400 cm^−1^, related to the vibration of the C-O bond of the (-CH-OH) group, and 1380 cm^−1^, associated with the C-O vibration of the (-CH_2_-OH) group, which were also different from expected and similar to those observed for the peaks at 1640, 1315, and 1545 cm^−1^. The band at 1380 cm^−1^ showed the most evident transmittance and shape changes, which could indicate stronger chemical interactions between these alcohol groups. These hydroxyl groups are the lateral branches of the main polymeric chain, and consequently, they are more available to form hydrogen bonds or other chemical interactions.

In the case of the composite materials made of HAp nanofibers, Figure 7c,d show the spectra of the pure HAp nanofibers, pure chitosan, and the different composite materials. The band produced by the phosphate groups in the HAp nanofibers appeared at 1026 cm^−1^, and it was narrower than that observed for commercial HAp (Figure 7a). This could indicate that the phosphate groups in the nanofibers were freer to vibrate, and many were closer to the nanofiber surface. When the composite materials were prepared, the band of phosphate groups that originally appeared at 1026 cm^−1^ had an evident shift up to a value of 1015 cm^−1^, for the sample with 5% by weight of nanofibers, and up to a wavenumber of 1000 cm^−1^, registered for the sample containing 10% by weight. However, for the sample containing 20% by weight of nanofibers, the band produced by the phosphate groups appeared again at 1015 cm^−1^. Moreover, the band produced by the phosphate groups showed consistent widening. The higher the HAp concentration in the composite material was, the wider the band was. The shift and widening of this band indicate their chemical interactions, and these variations were more evident than those observed for the band produced by the phosphate groups in the reference materials. The transmittance values of the bands also varied. The resulting transmittance of this band for each composite material was attributed to both the overlap of the bands at 1063 and 1048 cm^−1^, associated with the vibration of C-O bonds, and the chemical interaction of phosphate groups. The transmittance values consistently increased with the increase in HAp concentration in the composite material, which could indicate the chemical interactions formed by the phosphate groups.

In addition, for the composite materials made of HAp nanofibers, the bands located at 1640, 1315, and 1545 cm^−1^ had more relevant changes than those registered for the same bands in the reference materials. The band at 1640 cm^−1^ showed a small widening when the HAp concentration was increased from 5% to 10% by weight of nanofibers, and its transmittance decreased as expected, but this change in the transmittance was not as great as that observed for the same band in the reference materials made of commercial HAp. The peak at 1315 cm^−1^ also showed widening and transmittance changes. These variations indicated that the carbonyl groups of the amide groups of chitosan had chemical interactions.

The band at 1545 cm^−1^ also experienced a widening, as can be seen in Figure 7d, but the transmittance values of the samples with 5% and 10% by weight were similar. These changes in the band related to the vibrations of the N-H bonds of the amino groups were evidence of the chemical interactions between these functional groups.

The peaks at 1400 and 1380 cm^−1^ also showed a slight shift to lower wavenumbers, and the shape of the band at 1380 cm^−1^ in the materials made of nanofibers was more evident than those observed for the same band in the reference materials. This was evidence of the stronger chemical interactions between the hydroxyl groups in the composite materials made of nanofibers.

When the HAp nanofiber concentration was increased in the composite up to 20% by weight, the bands located at 1640, 1315, 1545, 1400, and 1380 cm^−1^ showed a remarkable decrease in transmittance, and despite the reduction in the concentration of the biopolymer, their registered transmittance values were closer to those observed for similar bands in pure chitosan (see Figure 7d). The latter indicates that in this sample, the chemical interactions were fewer than those observed in the samples with 5% and 10% by weight. It is possible that the chitosan matrix or HAp nanofibers had reached saturation. Comparing Figure 7b,d, it was possible to observe that the variations registered in the bands at 1640, 1315, 1400, 1545, and 1380 cm^−1^ for both types of composite materials were different, and all variations were more evident for the materials made of nanofibers. Consequently, it could be concluded that the chemical interactions were more abundant for these types of materials. Relevant information of the FTIR signals and their variations in the different materials is summarized in Table 1.

#### 3.2.4. Raman

The Raman analyses corroborated the results obtained by FTIR. For the reference materials, relevant variations in intensity were recorded for some peaks in the spectra. The broad signal at approximately 1650 cm^−1^ was produced by the contribution of both carbonyl groups C=O from the partially deacetylated amides and the amine groups [62,63,64]. This band remarkably changed in intensity. For pure chitosan, the intensity of this signal was rather visible, but when commercial HAp was incorporated, the band’s intensity was almost completely reduced, as can be observed in Figure 8a. In Reference 1, this peak was barely observable, and in Reference 2 and Reference 3, this signal almost disappeared. This reduction in the intensity of the peak was related to the chemical interactions between the carbonyl and amine groups. A similar variation was observed in other peaks located at 1115 cm^−1^, related to the C-O-C bond; 1080 and 1030 cm^−1^, produced by the C-O bond vibration; and 895 cm^−1^, corresponding to the C-O-C bond. All these peaks were produced by the chitosan matrix, which considerably reduced their intensity when the HAp content was increased to 5% by weight. This variation in the intensity was more evident when the HAp concentration was incremented up to 10%, but the intensity of these peaks did not vary by large amounts when the HAp concentration reached 20% by weight. A similar result was observed in FTIR analysis.

The peak that showed the most evident changes was that at 960 cm^−1^, related to the phosphate groups in HAp [65]. The signal varied in intensity and width. Reference 1 showed a smaller peak than that observed in pure HAp, as expected; however, in Reference 2, despite the increase in concentration, the intensity markedly decreased. Subsequently, the HAp concentration was increased again up to 20% in Reference 3, but the peak intensity increased less than expected, and the bandwidth was slightly reduced, as can be seen in Figure 8a. These variations could be attributed to the chemical interactions of the phosphate groups. The position of the band at 960 cm^−1^ did not vary, possibly because there were insufficient chemical interactions of the phosphate groups to produce a shift in the band.

Similar variations in the Raman spectra were detected for composite materials made of nanofibers. Once again, the signal at about 1650 cm^−1^ was observed for pure chitosan, but it almost disappeared for composite materials with different concentrations of HAp nanofibers, as shown in Figure 8b. The other peaks at 1115, 1080, 1030, and 895 cm^−1^ showed an expected reduction in the intensity when the HAp concentration was increased to 5% and 10% by weight, respectively, but the intensity of all these signals increased when the HAp content was increased up to 20%. The latter could be explained by saturation in the composite material, which reduced the chemical interactions of the involved functional groups. This assumption is supported by the observed changes in the signal at 960 cm^−1^ produced by the phosphate groups. The composite material with 5% by weight of HAp nanofibers showed an evident reduction in the intensity of this peak compared with that observed for the same band in the pure HAp nanofibers. The HAp concentration was increased to 10%, but despite this, the intensity of this band considerably decreased. The composite material exhibited the highest number of chemical interactions. When the HAp concentration was increased up to 20% by weight, the intensity of the band at 960 cm^−1^ was increased again, which suggests that the chemical interactions in this composite material reached saturation. The position of this band slightly varied, and the increment in the HAp nanofibers in the composite materials produced a small shift in the band at 960 cm^−1^ and moved it toward higher wavenumbers, as can be observed in Figure 8b. This variation indicated that there were a higher number of chemical interactions in the composite materials made of nanofibers. Important data of the Raman signals and their variations in the different materials are summarized in Table 2.

The FTIR and Raman results suggest that the phosphate groups of HAp (especially oxygen atoms) formed hydrogen bonds with the amine groups of chitosan. Other functional groups in the chitosan molecules that contained oxygen atoms, such as carbonyl C=O of the partially deacetylated amides, hydroxyl groups (-C-OH) and C-O-C bonds (particularly those oxygen atoms that bind glucose molecules), could form coordinate covalent bonds with the calcium ions of HAp because of the lone pairs of electrons. Although the Ca-O bond was not registered in the HAp–chitosan composite materials, this link has been observed in HAp–protein composite materials [66] or HAp–amino acid complexes [67], which is evidence of the chemical interactions between the carboxyl groups in the protein and the calcium ion in HAp. The lack of link registration may be due to the low number of interactions.

The particles of commercial HAp had different crystalline planes exposed on their surfaces; the most abundant were the (211) planes according to PDF #09-432, and these planes had few phosphate and calcium ions on their surfaces. According to the HAp crystal structure of the nanofibers [2] and visualizations of that structure using VESTA software, the most abundant planes, (300), had more calcium ions and phosphate groups on their surfaces than those on the (211) planes. This suggested that the nanofiber could form a greater number of chemical interactions with the chitosan matrix. Figure 9 shows some chemical interactions between the HAp nanofibers and chitosan molecules.

#### 3.2.5. Mechanical Properties

Tension tests were performed on both types of composite materials, and these experiments yielded relevant results regarding the mechanical resistance of different materials. Figure 10a shows the typical stress–strain curves for pure chitosan and the reference materials. Pure chitosan produced a stress–strain curve with a long elastic region and an extended plastic region, which are both characteristics of a highly deformable material. Pure chitosan reached an ultimate strength of approximately 107 MPa, and specimen fracture occurred almost immediately.

The addition of commercial HAp generated stiffness in the composite material. First, Reference 1 showed a decrease in Young’s modulus of approximately 40.6% compared with the pure chitosan specimen, and this reference material scarcely showed a plastic region and broke earlier than expected. When the HAp concentration was increased to 10% by weight (Reference 2), the composite material increased its Young’s modulus by 19.2% approximately, referred to as pure chitosan, but this material suffered a rupture with a low ultimate strength value 33% lower than that observed for the pure chitosan sample. Once again, the HAp content was increased to 20% by weight in Reference 3, but the mechanical resistance did not vary too much for this sample. The Young’s modulus was improved by approximately 21%, and the ultimate strength was 30% lower, related to the pure chitosan sample. Additionally, this latter sample showed remarkable debonding prior to fracture, which indicated that the organic and inorganic phases were separated due to the applied effort. The stress–strain curves displayed in Figure 10a show the above-mentioned stiffening of the composite materials, and they also show the considerable reduction in their plastic region produced by the increment in HAp concentration. In addition, the increment in mechanical resistance was notable from Reference 1 to Reference 2; however, in Reference 3, despite the increase in the HAp concentration, the mechanical resistance under tension was similar to that observed in Reference 2. The latter is correlated with the FTIR and Raman results; in fact, it was detected that Reference 3 generated fewer chemical interactions among the phosphate, amine, and carbonyl groups, which affected the mechanical properties of this material.

In the case of composite materials made of HAp nanofibers, remarkable differences in mechanical resistance were observed, as shown in Figure 10b. The most notable difference was the increment in the plastic region of all these composite materials compared with that observed in the reference materials. The composite material made with 5% by weight of nanofibers showed a decrease in Young’s modulus of 33.5% and a reduction in the ultimate strength of 16.6%, both related to the values registered for the pure chitosan sample. However, the plastic region of this material was similar to that observed for the pure biopolymer. At this concentration, the nanofibers did not stiffen the biopolymer.

The nanofiber concentration was increased to 10% by weight, and the Young’s modulus of this material increased by 22.2% relative to pure chitosan, with the ultimate strength also being increased by 29.2%. Finally, the nanofiber content was increased up to 20% by weight, and once again, there was a marked increase in the mechanical resistance under tension of this composite material. This material showed an increase in the Young’s modulus and in the ultimate strength of 52% compared with the values registered for pure chitosan. Despite the increase in the mechanical resistance of the last two materials (Figure 10b), it was possible to see a reduction in their plastic region compared with that observed in the pure biopolymer. This reduction in plasticity could be a consequence of HAp concentration. The FTIR and Raman results showed a significant increase in the number of chemical interactions in all the composite materials made of nanofibers. The increase in tenacity was evident in these materials, and this was attributable, in part, to the greater interrelations among the constituent molecules, which was caused by a large number of chemical interactions between the inorganic and organic phases.

The average values of Young’s modulus and ultimate strength of all composite materials are displayed in Table 3. In both types of composite materials (references and composite materials made of nanofibers), the tendency to increase the mechanical resistance under tension with the increase in the HAp concentration was evident. This tendency could be the result of the good dispersion of HAp aggregates in the chitosan matrix, which was produced by the use of a sonotrode to mix the components of the composite materials. From the data in Table 3, it can be said that the materials made of nanofibers showed a higher mechanical resistance under tension than the reference materials.

The observations of the fracture profiles using SEM corroborated the results obtained in the tensile tests and by FTIR. Figure 11 shows the micrographs of the fracture surfaces produced in the HAp samples after the tensile tests. In both cases, the dispersion of HAp aggregates within the polymer matrix was quite good because of the use of the sonotrode (see Figure 11a,c). Even though both types of HAp nanostructures (particles and nanofibers) presented a large number of chemical interactions according to the FTIR results, the nanofibers remained attached to the biopolymer matrix after the tensile test, and they broke owing to the applied effort, as shown in Figure 11d. Finally, the nanofibers contributed with their resistance to the final mechanical properties of the composite material. In contrast, the commercial HAp particles became unstuck during the mechanical test, and their contribution to the final resistance of the composite material was poor (Figure 11b).

## 4. Discussion

The improvement of the mechanical resistance of the composite materials made of fibers was the result of using HAp with a nanofiber morphology. The length of these fibers made them act as a mechanical reinforcement. A fiber can resist a tensile stress better than a particle. Moreover, the nanofibers were grown along the “c” axis of the HAp crystal structure [2], with a structure similar to tooth enamel and also having a similar mechanical resistance [68,69,70]. The high purity and high crystallinity of the HAp, the arrangement of nanofibers within microfibers with a hexagonal cross-section [45,46,47], and the preferential crystalline orientation were the main factors that contributed to improving the mechanical properties of the composite materials.

The lateral surfaces of the fibers were very flat, and they were formed by the (100) and (300) crystalline planes, which contained several phosphate groups and calcium ions; as it was expected, these cations and anions produced a sufficient number of chemical interactions with the chitosan matrix, which generated synergistic mechanical properties.

The sonotrode used for the synthesizing of the composite materials provided enough energy to homogeneously disperse the HAp nanofibers with no considerable damage to their structure, conserving their mechanical properties.

## 5. Conclusions

The synthesis of HAp/Chitosan composite materials was successful. The use of a sonotrode was effective in achieving the good dispersion of HAp aggregates within the biopolymer matrix, which had a positive impact on the final mechanical resistance under tension of the materials.

The addition of HAp nanofibers to the chitosan matrix was critical for increasing the mechanical resistance of the composite materials. The crystalline and structural characteristics of the HAp nanofibers contributed to achieving better mechanical properties under tension of the material made with them. The mechanical resistance under tension of these materials was similar to that reported for cortical bone; therefore, these composite materials have potential applications as implant materials for bone tissue.

## Figures and Tables

**Figure 1 materials-15-04718-f001:**
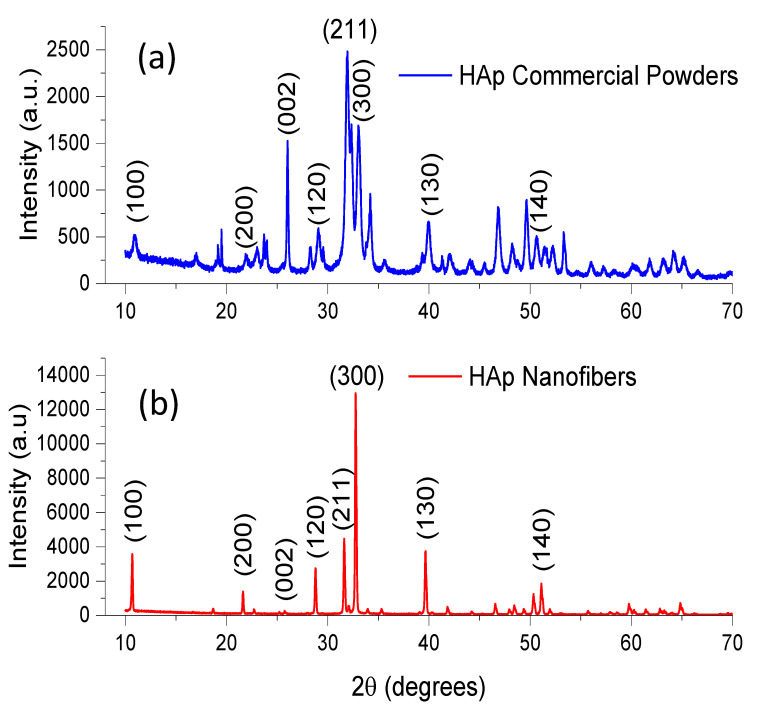
Typical XRD diffractograms of (**a**) commercial HAp powders and (**b**) HAp nanofibers. Notice the preferential crystalline orientation in the [001] direction. Indexing is indicated in both cases.

**Figure 2 materials-15-04718-f002:**
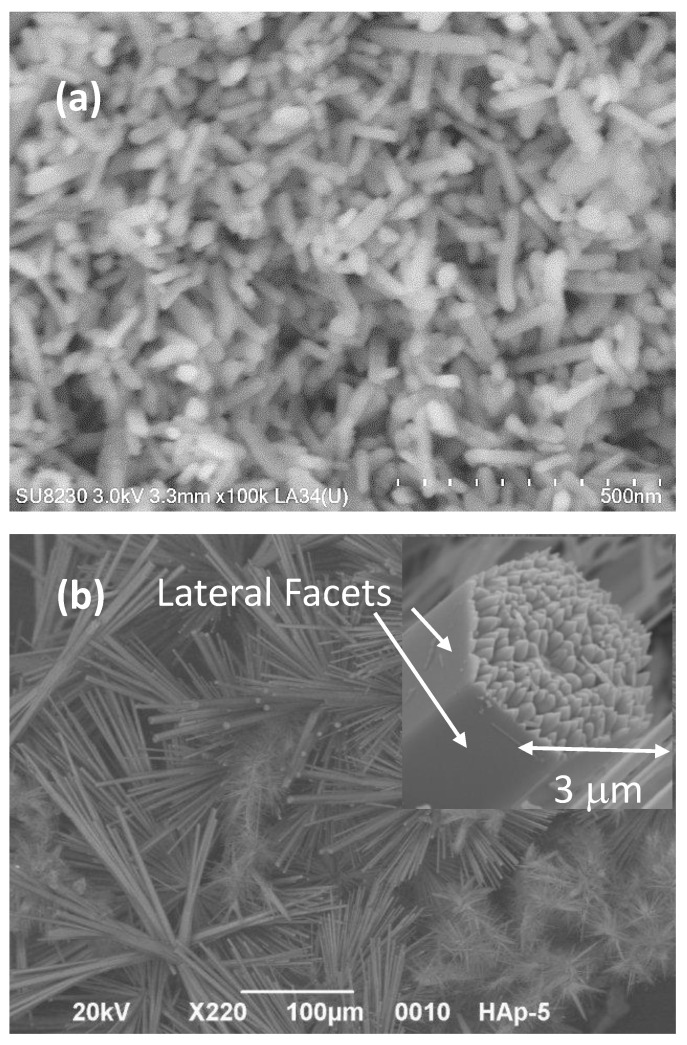
Secondary-electron micrographs obtained by SEM of (**a**) commercial HAp powders and (**b**) HAp nanofibers.

**Figure 3 materials-15-04718-f003:**
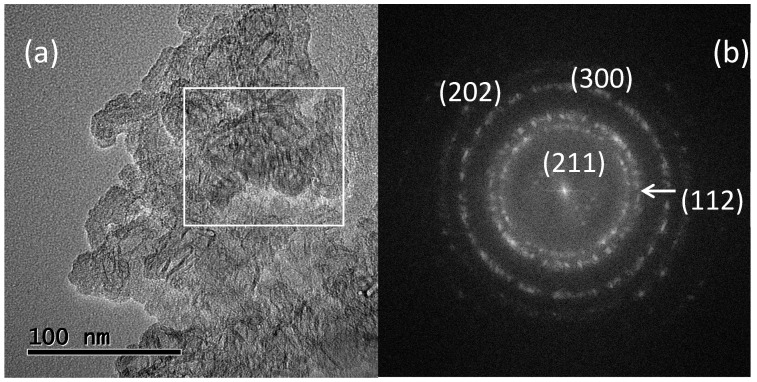
(**a**) HRTEM micrograph of HAp commercial powders. Atomic resolution can be seen. (**b**) FFT applied to the image region within the white rectangle produced a ring pattern. The FFT was indexed.

**Figure 4 materials-15-04718-f004:**
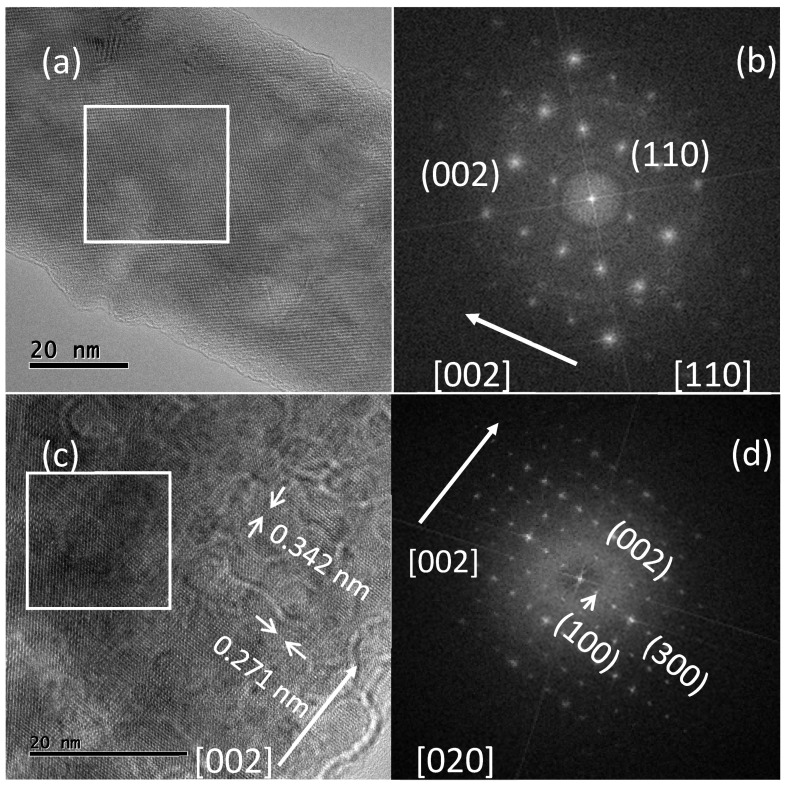
HRTEM micrographs of different HAp nanofibers (**a**,**c**). (**b**) Dot pattern produced by the image region in the white rectangle in (**a**) when a FFT was applied, and (**d**) other dot pattern generated by the image region inside the rectangle in (**b**). Both dot patterns are indexed and the white arrows indicate the direction of fiber growth.

**Figure 5 materials-15-04718-f005:**
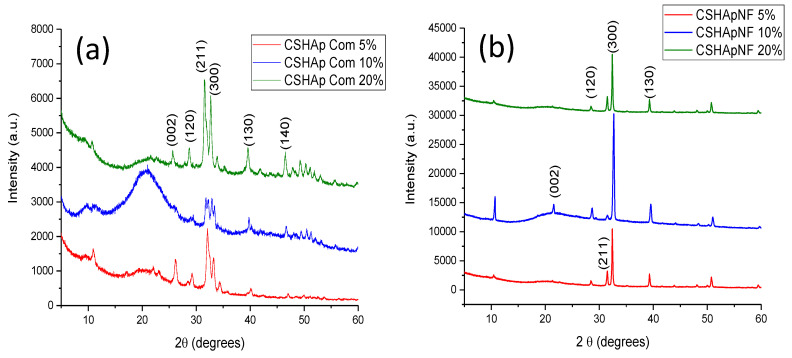
XRD diffractograms of (**a**) reference materials and (**b**) composite materials made of nanofibers. In both cases, the main Bragg reflections are indexed.

**Figure 6 materials-15-04718-f006:**
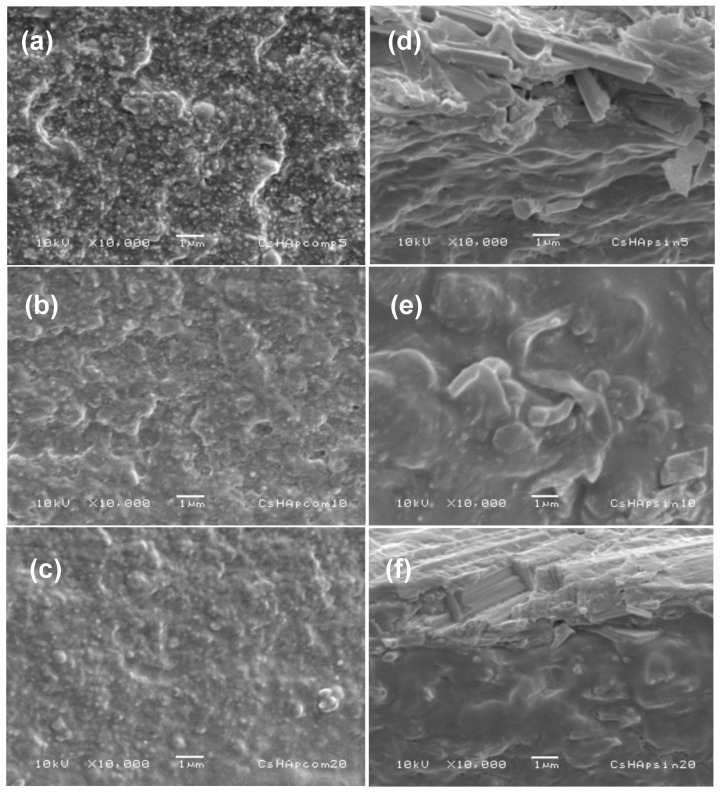
SEM secondary-electron micrographs of reference materials (**a**–**c**) and of HAp nanofibers/chitosan composite materials (**d**–**f**).

**Figure 7 materials-15-04718-f007:**
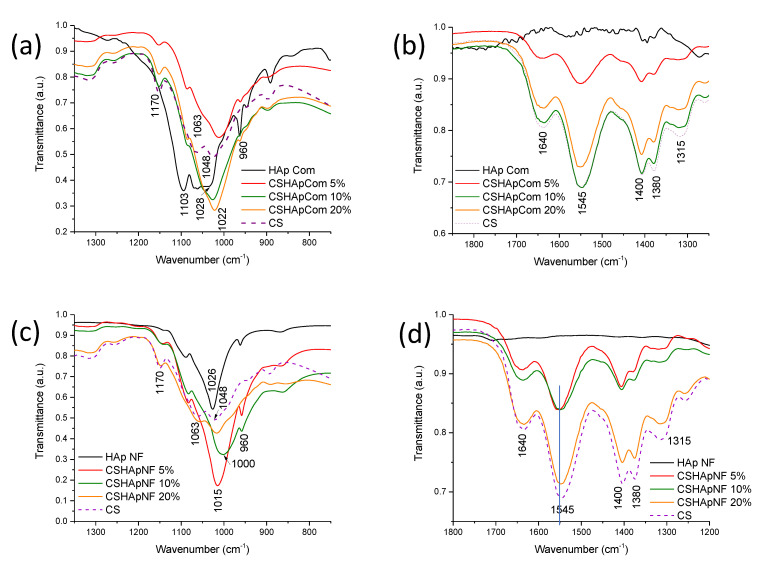
FTIR spectra of reference materials (**a**,**b**), and HAp nanofibers/chitosan composite materials (**c**,**d**).

**Figure 8 materials-15-04718-f008:**
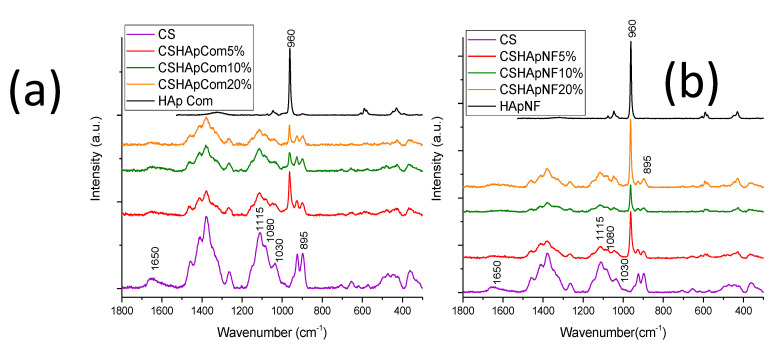
Raman spectra of (**a**) reference materials and (**b**) HAp nanofibers/chitosan composite materials.

**Figure 9 materials-15-04718-f009:**
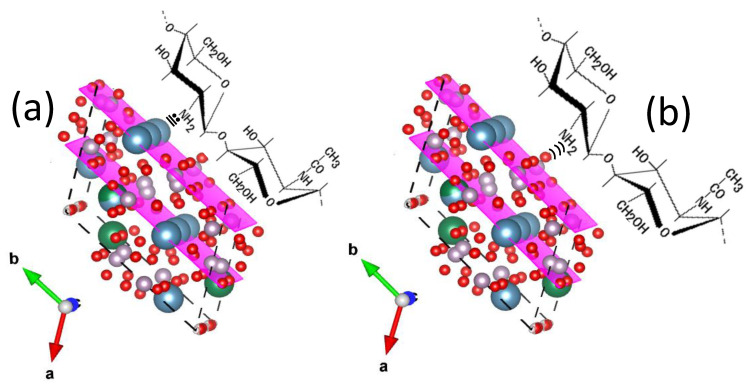
Schematic representations of possible chemical interactions between (**a**) calcium ion in HAp and amine groups in chitosan molecules, and (**b**) hydrogens in amine groups of chitosan and oxygen atoms in phosphate groups in HAp. The (300) planes were used for this representation.

**Figure 10 materials-15-04718-f010:**
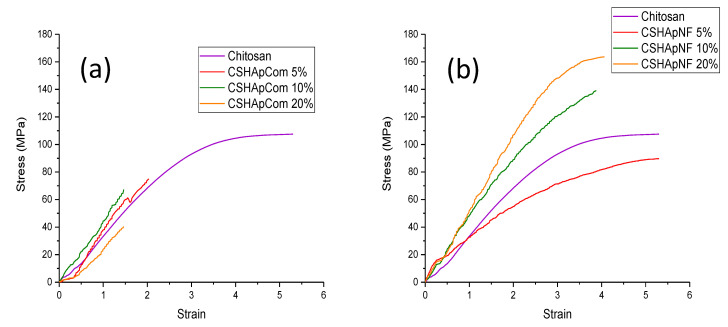
Stress–strain curves for (**a**) reference materials and (**b**) HAp nanofibers/chitosan composite materials. Notice the differences in mechanical resistance under tension.

**Figure 11 materials-15-04718-f011:**
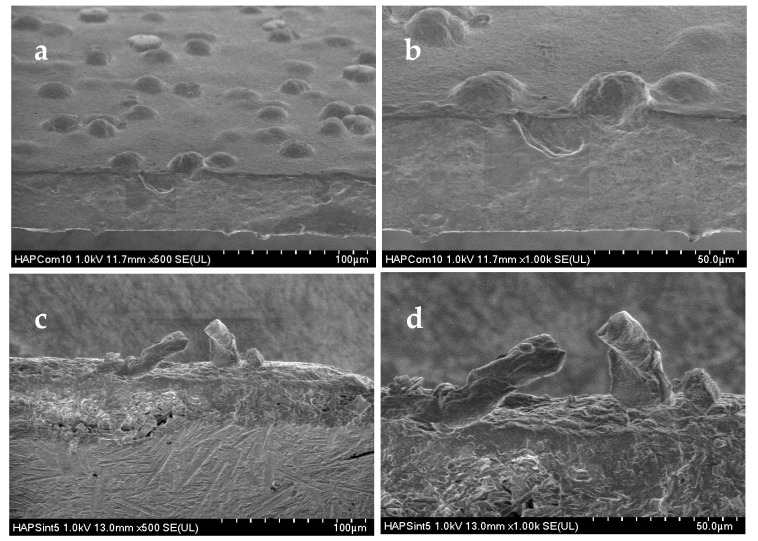
SEM micrographs of the fracture surfaces produced by tensile tests on the reference material with 20% by weight of commercial HAp (**a**,**b**) and composite material made of 20% by weight of nanofibers (**c**,**d**). Notice the HAp nanofibers attached to the polymer matrix on the fracture surface.

**Table 1 materials-15-04718-t001:** Positions of and variations in FTIR signals from all types of materials.

Signal at (cm^−1^) in Pure HAp or Pure Chitosan	Group	Type of Vibration	Signal at (cm^−1^) in Composite Material and Type of Variation in the Signal	References
960	Phosphate	Symmetrical deformation	960; transmittance variation	[58,59]
1103	Phosphate	Asymmetrical deformation	1130; transmittance variations	
1028 Commercial HAp	Phosphate		Position shift to 1022 RM 1 1024 RM 2 and 3	
1026 HAp Nanofibers (narrower than comm. HAP)	Phosphate	Asymmetrical deformation	Consistent widening and position shift to: 1015 in CNF with 5% NF; 1000 in CNF with 10% NF; 1015 in CNF with 20% NF	
1048	-C-O	Vibration of glucose ring	Transmittance variations in RM Overlap with signal at 1026 in CNF	[55,56,58,59]
1063				
1315	C=O	Symmetrical vibration	1315; transmittance variation in RM; widening and transmittance changes in CNF	[60]
1380	C-O	Vibration of (-CH_2_-OH) group	1380; unexpected transmittance variations in RM; shift in CNF	[61]
1400	C-O	Vibration of (-CH-OH) group	1400; unexpected transmittance variations in RM; shift in CNF	
1545	-N-H	Vibration of amines	1545; unexpected transmittance variations in RM; widening in CNF	
1640	C=O	Strong vibration of amides	1640; transmittance variation in RM; widening and expected transmittance variation in CNF	[60]

RM = reference materials, CNF = composite materials with nanofibers.

**Table 2 materials-15-04718-t002:** Positions of and variations in Raman signals from all types of materials.

Signal at (cm^−1^) in Pure HAp or Pure Chitosan	Group	Type of Vibration	Signal at (cm^−1^) in Composite Material and Type of Variation of the Signal	References
895	C-O-C	Symmetric stretch	895; irregular intensity reduction in RM 895; expected intensity reduction in CNF	[27,57,65]
1030	C-O	Deformation	1030; irregular intensity reduction in RM 1030; expected intensity reduction in CNF	
1080			1080; irregular intensity reduction in RM 1080; expected intensity reduction in CNF	
1115	C-O-C	Asymmetric stretch	1030; irregular intensity reduction in RM 1030; expected intensity reduction in CNF	
960	Phosphate	Symmetrical deformation	960; intensity and width variation in RM 960; inconsistent intensity reduction	

RM = reference materials, CNF = composite materials with nanofibers.

**Table 3 materials-15-04718-t003:** Young’s modulus and ultimate strength average values obtained for all composite materials synthesized in this research study.

Sample	Young’s Modulus (MPa)	Ultimate Strength (MPa)
Reference 1	21.1927 ± 1.102	40.316 ± 2.177
Reference 2	42.5532 ± 2.16	71.5969 ± 3.56
Reference 3	43.2241 ± 2.171	74.6025 ± 3.954
CSHApNF 5%	23.7201 ± 1.138	89.664 ± 3.855
CSHApNF 10%	43.611 ± 1.831	138.881 ± 5.833
CSHApNF 20%	54.345 ± 2.44	163.603 ± 6.567
Chitosan	35.6892 ± 1.788	107.5055 ± 5.396

## Data Availability

Data are contained within the article.

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
