# Peer review of "Improving the Mechanical Resistance of Hydroxyapatite/Chitosan Composite Materials Made of Nanofibers with Crystalline Preferential Orientation"

_materials, 2022, doi:10.3390/ma15134718_

Round 1

Reviewer 1 Report

Here, the  stability and mechanical properties of hydroxyapatite (HAp)/Chitosan nanofibers were studied by mixing with commercial chitosan using a sonotrode to improve their dispersion within the biopolymer matrix and prevent their migration. The HAp nanofiber/Chitosan composite materials showed a higher mechanical resistance than that observed for similar materials with the same chemical composition and that were made of commercial HAp powders, which were used as reference materials. The introduction provide sufficient background and include all relevant references, the methods adequately described and the conclusions supported by the results. However, It is necessary to better reference the methodologies used in this work: 2.2.1 X-ray diffraction; 2.2.2 Scanning electron microscope; 2.2.3 High resolution electron microscopy; 2.3.1 X-ray diffraction; 2.3.2 Scanning electron microscopy; 2.3.3 Fourier transform infrared spectroscopy (FTIR); 2.3.4 Raman spectroscopy; 2.3.5 Mechanical Properties. Insert bibliographic references in the presentation of Results, when citing the literature. Same thing for Discussion.

Author Response

It is necessary to better reference the methodologies used in this work: 2.2.1 X-ray diffraction; 2.2.2 Scanning electron microscope; 2.2.3 High resolution electron microscopy; 2.3.1 X-ray diffraction; 2.3.2 Scanning electron microscopy; 2.3.3 Fourier transform infrared spectroscopy (FTIR); 2.3.4 Raman spectroscopy; 2.3.5 Mechanical Properties.

Answer: Done, references in this sections were included

Insert bibliographic references in the presentation of Results, when citing the literature. Same thing for Discussion.

Answer: Done, references in Results and discussion sections were included.

Reviewer 2 Report

The presented manuscript deals with improving mechanical resistance of HAp/Chitosan composite materials made of hexagonal cross-sectioned nanofibers with crystalline preferential orientation in which the hexagonal cross-sectioned HAp nanofibers were produced using a microwave-assisted hydrothermal method. Furthermore, the morphology and  mechanical resistance of the composite materials were also discussed. The manuscript was written well. However some comments need to be incorporated before publication.

Please revise the title of the manuscript as it is very lengthy and avoid the use of short form (HAp).

Please revise the abstract by introducing the results (in term of values).

In introduction section, literature review should be extended and up to dated using the suggested ref. (Polymers 14 (4), 845 and Journal of Cleaner Production 318, 128603). Also mention why the author choose chitosan.

Please check the typo and grammetical mistakes thoughout the manuscript (line 138, 143,144,264, 283 etc)

In my opinion, this is not a wise decision to separate the result of  the characterization (HAp Nanofibers and composite material) as it increase the length of article and decrease the interest of readers.

why the authors did not perform the TEM analysis in case of composite material.

Some figure need to improve in respect of quality like figure 2a and 9.

Discussion and conclusion part are very undersized.

Author Response

Please revise the title of the manuscript as it is very lengthy and avoid the use of short form (HAp).

Answer: Manuscript title was revised and modified.

Please revise the abstract by introducing the results (in term of values).

Answer: abstract is limited to 200 words; therefore, no results values were mentioned in this section of the manuscript.

In introduction section, literature review should be extended and up to dated using the suggested ref. (Polymers 14 (4), 845 and Journal of Cleaner Production 318, 128603).

Answer: Done, all references in the introduction section were revised and updated.

Also mention why the author choose chitosan.

Answer: In the introduction section is now mentioned why chitosan was selected

Please check the typo and grammetical mistakes thoughout the manuscript (line 138, 143,144,264, 283 etc)

Answer: An exhaustive revision of the English was done in the whole manuscript

In my opinion, this is not a wise decision to separate the result of the characterization (HAp Nanofibers and composite material) as it increase the length of article and decrease the interest of readers.

Answer: The authors decided to separate the characterization of HAp and the composite materials to emphasize the HAp crystal structure examination for those readers whose area of interest is crystallography

why the authors did not perform the TEM analysis in case of composite material.

Answer: In fact, it was tried to do. The observation of a thin film of HAp-chitosan was difficult due to the rapid degradation of this material under the electron beam

Some figure need to improve in respect of quality like figure 2a and 9.

Answer: Contrast was enhanced to figure 2a. The figure has enough resolution, but the sample was electric charged and not good images were collected. Figure 9 was taken directly from the Vesta software and its quality could not be improved.

Discussion and conclusion part are very undersized

Answer: In the results section all characterization results were analyzed and discussed in detail. Discussion and conclusion sections showed a summary of all that information. A more detailed analysis in those sections could be redundant. 

Reviewer 3 Report

This is a very good article, however, this version does not look worthy and cannot be recommended for publication in this form and at least needs some improvement and clarification.

1.     Introduction.  Almost all of the first 20 references are older than 10 years. Because of this, the motivation and relevance of this study may be in doubt, because the latest achievements in HAP research and development are not reflected.  This information can be updated, using the search for the latest achievements: https://www.mdpi.com/search?q=hydroxyapatite

2.     Introduction. Line 40. it is necessary to note their resistance to both ultraviolet and X- irradiations. The outcome of HAP indeed depends on their resistance to aging, including radiation. See:

Bystrova, A.; Dekhtyar, Y.D.; Popov, A.; Coutinho, J.; Bystrov, V. Modified hydroxyapatite structure and properties: Modeling and synchrotron data analysis of modified hydroxyapatite structure. Ferroelectrics 2015475, 135–147.

Hübner, W.; Blume, A.; Pushnjakova, R.; Dekhtyar, Y.; Hein, H.-J. The influence of X-ray radiation on the mineral/organic matrix interaction of bone tissue: An FT-IR microscopic investigation. Int. J. Artif. Organs 200528, 66–73.

3.     Data on Fig. 7 are very important, but there is no detailed comparison with literature data. It is recommended to supplement this paragraph with a table that lists all the peaks, their interpretation, and a couple of columns with literature data. This table will be very useful and informative for future potential readers.

4.     The same is for Raman spectra data.

5.     Please check the following references as there are no volume or page numbers: 22,35,38, 39,43, 44, 45, 46,47,48, 49, 50, 51,52.

Author Response

  1. Almost all of the first 20 references are older than 10 years. Because of this, the motivation and relevance of this study may be in doubt, because the latest achievements in HAP research and development are not reflected.  This information can be updated, using the search for the latest achievements: https://www.mdpi.com/search?q=hydroxyapatite

Answer: Done, all references in the introduction section were revised and updated. Reference 2, although it is old, contains relevant information for the analysis of the HAp crystal structure.

  1. Introduction. Line 40. it is necessary to note their resistance to both ultraviolet and X- irradiations. The outcome of HAP indeed depends on their resistance to aging, including radiation. See:

Bystrova, A.; Dekhtyar, Y.D.; Popov, A.; Coutinho, J.; Bystrov, V. Modified hydroxyapatite structure and properties: Modeling and synchrotron data analysis of modified hydroxyapatite structure. Ferroelectrics 2015475, 135–147.

Hübner, W.; Blume, A.; Pushnjakova, R.; Dekhtyar, Y.; Hein, H.-J. The influence of X-ray radiation on the mineral/organic matrix interaction of bone tissue: An FT-IR microscopic investigation. Int. J. Artif. Organs 200528, 66–73.

Answer: Suggested literature was revised and were included in the reference list

  1. Data on Fig. 7 are very important, but there is no detailed comparison with literature data. It is recommended to supplement this paragraph with a table that lists all the peaks, their interpretation, and a couple of columns with literature data. This table will be very useful and informative for future potential readers.

Answer: Table 1 was added to manuscript text containing relevant information of FTIR signals and some references

  1. The same is for Raman spectra data.

Answer: Table 2 was added to manuscript text containing relevant information of Raman signals and some references

  1. Please check the following references as there are no volume or page numbers: 22,35,38, 39,43, 44, 45, 46,47,48, 49, 50, 51,52.

Answer: the mentioned references were revised and the missing information was added.

Round 2

Reviewer 3 Report

The authors have significantly improved the manuscript, now it can be recommended for publication.